# Characterization of IgG Antibody Response against SARS-CoV-2 (COVID-19) in the Cypriot Population

**DOI:** 10.3390/microorganisms10010085

**Published:** 2021-12-31

**Authors:** George Krashias, Elie Deeba, Astero Constantinou, Maria Hadjiagapiou, Dana Koptides, Jan Richter, Christina Tryfonos, Stavros Bashiardes, Anastasia Lambrianides, Maria A. Loizidou, Andreas Hadjisavvas, Mihalis I. Panayiotidis, Christina Christodoulou

**Affiliations:** 1Cyprus School of Molecular Medicine, Nicosia 2371, Cyprus; Mariah@cing.ac.cy (M.H.); nancyl@cing.ac.cy (A.L.); loizidou@cing.ac.cy (M.A.L.); ahsavvas@cing.ac.cy (A.H.); mihalisp@cing.ac.cy (M.I.P.); cchristo@cing.ac.cy (C.C.); 2Department of Molecular Virology, The Cyprus Institute of Neurology and Genetics, Nicosia 2371, Cyprus; elied@cing.ac.cy (E.D.); astero@cing.ac.cy (A.C.); dankom@cing.ac.cy (D.K.); richter@cing.ac.cy (J.R.); tryfonos@cing.ac.cy (C.T.); sbash@cing.ac.cy (S.B.); 3Department of Neuroimmunology, The Cyprus Institute of Neurology and Genetics, Nicosia 2371, Cyprus; 4Department of Cancer Genetics, Therapeutics and Ultrastructural Pathology, The Cyprus Institute of Neurology and Genetics, Nicosia 2371, Cyprus

**Keywords:** SARS-CoV-2, nucleocapsid (N) protein, Spike 1 receptor binding protein, IgG antibody, seroprevalence

## Abstract

The severe acute respiratory syndrome coronavirus 2 (SARS-CoV-2) pandemic has hit its second year and continues to damage lives and livelihoods across the globe. There continues to be a global effort to present serological data on SARS-CoV-2 antibodies in different individuals. As such, this study aimed to characterize the seroprevalence of SARS-CoV-2 antibodies in the Cypriot population for the first time since the pandemic started. Our results show that a majority of people infected with SARS-CoV-2 developed IgG antibodies against the virus, whether anti-NP, anti-S1RBD, or both, at least 20 days after their infection. Additionally, the percentage of people with at least one antibody against SARS-CoV-2 in the group of volunteers deemed SARS-CoV-2 negative via RT-PCR or who remain untested/undetermined (14.43%) is comparable to other reported percentages worldwide, ranging anywhere from 0.2% to 24%. We postulate that these percentages reflect the underreporting of true infections in the population, and also show the steady increase of herd immunity. Additionally, we showed a significantly marked decrease in anti-NP IgG antibodies in contrast to relatively stable levels of anti-S1RBD IgG antibodies in previously infected individuals across time.

## 1. Introduction

Severe acute respiratory syndrome coronavirus 2 (SARS-CoV-2), responsible for coronavirus disease 2019 (COVID-19), has already caused a loss of over 4.6 million lives globally (13 September 2021) [1].

The Republic of Cyprus recorded its first COVID-19 case in March 2020 in Nicosia. After two months of soaring COVID-19 cases, numbers regressed up to October 2020, when the first large wave of COVID-19 was recorded [1]. This first COVID-19 wave was attributed to the B.1.258 linage that spread rapidly and largely dominated the autumn/winter seasons, with a peak lineage prevalence of 86% among SARS-CoV-2 infected individuals in the Republic of Cyprus during the months of November 2020 and December 2020 [2]. Since then, two additional large COVID-19 waves have been recorded, in February–May 2021 and June–September 2021. The Alpha (B.1.1.7) and Delta (B.1.617.2) variants have been primarily responsible for driving the second and third COVID-19 waves, respectively.

Since the beginning of the COVID-19 pandemic, Cyprus has been amongst the countries characterized by extensive SARS-CoV-2 testing (RT-PCR and rapid antigen test) [3]. As of September 2021, Cyprus has conducted 58.95 daily tests per thousand, which is considered to be the highest at a global scale [4]. Of equal importance, Cyprus has 23.568 daily confirmed cases per hundred thousand [4].

Two years into the SARS-CoV-2 pandemic, the official global number of reported cases of the disease remains largely underestimated. Back in May 2020, it was extrapolated that the prevalence and transmissibility of SARS-CoV-2 is essentially underreported [5], considering the proportion of cases that go unrecognized, i.e., asymptomatically infected individuals, individuals with flu-like symptoms, or individuals who do not test in the first place. Indeed, although we have witnessed a constant fluctuation of reported cases throughout the past seasons, the SARS-CoV-2 pandemic does not seem to be coming to a stop. In fact, on 13 September 2021, there was a cumulative total of about 225.27 million confirmed cases of COVID-19 worldwide since the start of the pandemic [6].

For such reasons, seroprevalence studies are currently being conducted worldwide to supplement the data on people exposed/infected with SARS-CoV-2, shedding light on the heterogeneity of the symptoms associated with infection as well as disease severity. Reports are being published, sometimes on a weekly basis, of seroprevalence on a country-wide, regional, and even local basis. In Greece, for instance, one country-wide study placed seroprevalence at 0.36% between March and April of 2020 [7], whereas sub-local studies estimated an averaged range of 0.93% to 2.18% between April and July of 2020 [8,9,10]. A SeroTracker tool has been constructed to monitor worldwide reports of SARS-CoV-2 serological data and create a SARS-CoV-2 serosurveillance database platform [11]. Of equal importance, monitoring of SARS-CoV-2-specific antibody responses over time provides a valuable tool to closely monitor waning immunity, contributing at the same time to the ongoing governmental preparedness to combat the SARS-CoV-2 pandemic. Such population studies are by far the most effective way to gain knowledge about the prevalence of asymptomatic or mildly symptomatic cases, which contribute to the spread and sustainability of the virus without being subject to the same chain tracking as that of the classical symptomatic cases [12].

To the best of our knowledge, there is no information related to the percentage of infected individuals who were able to produce antibodies against SARS-CoV-2 in Cyprus. Of equal importance, serological data in subjects with no history of SARS-CoV-2 infection are still lacking. In the spirit of this global effort, we aimed to characterize the seroprevalence of SARS-CoV-2 antibodies in the Cypriot population for the first time since the pandemic started. We also aimed to measure the progression of SARS-CoV-2 antibody levels in SARS-CoV-2-infected individuals across time as a means of monitoring their antibody-mediated immunity after a SARS-CoV-2 infection.

## 2. Methodology

### 2.1. Ethical Approval and Subject Recruitment

This study was approved by the Cyprus National Bioethics Committee (ΕΕΒΚ/ΕΠ/2020/23). The study inclusion criteria were >18 years of age, with either a positive or negative SARS-CoV-2 RT-PCR detection test. Study participants were recruited following advertisement on social media and television. Upon enrolment, all participants provided information related to previous history of COVID-19, i.e., either RT-PCR or rapid antigen test results for SARS-CoV-2 detection. All volunteers completed and signed an informed consent form.

### 2.2. Study Population and Sample Collection

According to the statistical service of the Republic of Cyprus, the latest population recording was at 888,000 at the end of 2019 [13]. A total of 889 volunteers signed up for this study, which corresponds to 0.1% of the total population. Note that the majority of the volunteers were residents of the city of Nicosia. Of these 889 volunteers, 695 were positive for SARS-CoV-2 infection either via rapid antigen test or RT-PCR confirmation and 194 were SARS-CoV-2 negative via RT-PCR or remained untested/undetermined. Both groups were sex- and age-matched. Blood samples were collected from SARS-CoV-2-positive volunteers (PosV) between 20 days and 6 months after having reported positive (median [interquartile range]: 64 days [44 to 98.5]). Blood samples from SARS-CoV-2-negative/-unknown volunteers (NegV) were collected at periods matching those of PosV sampling. The sampling period was performed between October 2020 and May 2021. Our ability to take additional sequential samples was hindered by the advent of the SARS-CoV-2 vaccinations. Therefore, we relied on volunteers who had not yet been vaccinated and therefore, a second matched sample was taken from 205 antibody-positive volunteers 2 months and up to 6 months after the first sample (median [interquartile range]: 94 days [91 to 99]). These volunteers were chosen from either the PosV or NegV groups mentioned previously, given the condition that they have tested positive for the presence of both antibodies during the first sampling. Similarly, a third matched sample was obtained from 21 antibody-positive volunteers within a similar time frame (median [interquartile range]: 96 days [90 to 99]). Note that no SARS-CoV-2 detection retests were performed between the 3 sampling time points. The characteristics for all participants are summarized in Table 1.

Blood samples were collected in tubes containing clotting activators at the COVID-19 sampling unit of The Cyprus Institute of Neurology and Genetics. Following blood collection, samples were centrifuged for 10 min at 500× *g* at 20 °C to obtain cell-free serum. The serum was stored at −20 °C until analysis.

### 2.3. Sample Processing and Analysis

Serum obtained from the two groups of the study was used to qualitatively determine the positivity index (P.I.) of nucleocapsid protein (NP) and S1 receptor-binding domain (S1RBD) IgG antibodies. Anti-NP IgG was measured using a SARS-CoV-2 NP IgG ELISA kit (CE-IVD) (SKU 41A222, ImmunoDiagnostics Ltd., Hong Kong, China), and anti-S1RBD IgG was measured using a SARS-CoV-2 S1RBD IgG ELISA kit (CE-IVD) (SKU 41A235, ImmunoDiagnostics Ltd., Hong Kong, China). The positive controls of each ELISA were provided separately by the company as follows: humanized IgG monoclonal antibody against SARS-CoV-2 Nucleocapsid Protein ELISA kit (SKU 41A227, Immunodiagnostics Ltd., Hong Kong, China) and humanized anti-S1RBD IgG monoclonal antibody ELISA kit (SKU 41A236, Immunodiagnostics Ltd., Hong Kong, China). The absorbances of the samples were measured at 450 nm. Note that a summary of the protocol can be found in the Appendix A. Following the manufacturer’s protocol, the P.I. for the NP IgG was calculated as the ratio of the blanked absorbance of the sample acquired over 0.2. The P.I. for the S1RBD IgG was calculated as the ratio of the absorbance of the sample acquired to the cut-off absorbance value obtained from the cut-off sample provided by the kit. According to the manufacturer, values greater than 1.1 were considered positive for the respective antibody tested among groups. In addition, according to the manufacturer, the NP IgG ELISA kit had a sensitivity of 97% and a specificity of 99%, whereas the S1RBD IgG ELISA kit had a sensitivity of 92.5% and a specificity of 93.3%. Based on this information, the prevalence of each antibody in the groups tested was adjusted based on recommendations from Sempos et al. [14]. We kindly note that performing such adjustments “helps harmonize study results within countries and worldwide” and may lead to more accurate prevalence estimates even among differing antibody-measuring kits [14].

### 2.4. Statistical Analysis

The non-parametric Mann–Whitney U test was used to evaluate significance of the P.I. levels of the different antibodies tested among the SARS-CoV-2-positive volunteers and the SARS-CoV-2-negative/-unknown volunteers. The Fisher’s exact test was used to evaluate the significance of antibody presence among the study groups. The Wilcoxon signed-rank test was employed to evaluate the significance in the change between the first and second samplings of the positive volunteers, and the Friedman test was to evaluate the significance in the change between the first, second, and third samplings of the positive volunteers. The GraphPad Prism v8.00 for Windows software program was used to perform the statistical analyses (GraphPad Software, La Jolla, CA, USA).

## 3. Results

### 3.1. Seroprevalence of SARS-CoV-2 IgG in Previously Infected and Negative/Unknown Volunteers

Overall, NP-specific and S1RBD-specific IgG responses were detected in both groups, although at different frequencies (Table 2). In detail, IgG antibodies against NP were detected in 598 out of 695 PosV (adjusted prevalence; 88.59%), compared to 25 out of 194 NegV (adjusted prevalence; 12.38%) (*p* < 0.0001). Similarly, 645 out of 695 PosV (adjusted prevalence; 100%) tested positive for IgG antibodies against S1RBD, compared to 24 out of 194 NegV (adjusted prevalence; 6.61%) (*p* < 0.0001). Volunteers with at least one positive IgG (either positive for anti-NP or anti-S1RBD, or both) constituted 95.40% of the PosV group and 14.43% of the NegV group. Table 2 presents the crude and adjusted prevalence of each antibody in the groups tested.

Evaluating the level of antibodies in each group revealed that the median anti-NP IgG index was significantly higher (*p* < 0.0001) in PosV (median [interquartile range]: 7.415 [2.340 to 16.380]) than in NegV (0.150 [0.000 to 0.402]) (Figure 1). The median anti-S1RBD IgG index was similarly significantly higher (*p* < 0.0001) in PosV (6.029 [2.728 to 13.870]) than in NegV (0.380 [0.035 to 0.585]) (Figure 2).

### 3.2. Change in SARS-CoV-2 IgG Levels of Previously Infected Volunteers across Time

We acquired matched subsequent samples from antibody-positive volunteers, and evaluated the change in anti-NP and anti-S1RBD IgG antibodies. There was a significant drop in the P.I. of anti-NP IgG by 50.140% (*p* < 0.0001) in the 205 positive volunteers tested between their first and second sample (Figure 3A). Similarly, a significant decrease in the P.I. of anti-S1RBD IgG by 24.070% (*p* < 0.0001) was observed in that same group (Figure 3B).

A third sample was obtained from 21 antibody-positive volunteers, and the overall change across the three time points was evaluated. The decrease in P.I. of anti-NP IgG was found significant across the three time points, dropping by 10.090% between the first and second sampling and by 35.680% between the second and third samplings (*p* < 0.01) (Figure 4A). In contrast, the P.I. decrease of anti-S1RBD IgG was not significant across time, dropping by a total of 13.990% between the first and third samplings (Figure 4B).

## 4. Discussion

The current study contributes serological data pertaining to SARS-CoV-2 infection in the Cypriot population for the first time.

A majority of people infected with SARS-CoV-2 developed IgG antibodies against the virus, whether anti-NP, anti-S1RBD, or both, at least 20 days after their infection. This phenomenon confirms the presence of at least a humoral immune response against SARS-CoV-2 regardless of the demographic, as seen globally [15,16,17,18,19,20]. This observation is in agreement with the high seroconversion rates previously reported in large studies [21,22]. Nevertheless, a minority of people showed a negative result for SARS-CoV-2-specific antibodies. This can possibly be traced back to these individuals already having undetectable levels of antibodies reactive to the S1RBD and N proteins, or that enough time had passed between their infection and the sampling that antibodies were no longer within a detectable range. On the other hand, the presence of antibodies in volunteers who were not confirmed to be positive by RT-PCR could reflect the portion of the population that was asymptomatic or at the very least pauci-symptomatic and did not require further investigation into the nature of their illness. As a matter of fact, the percentage of people with at least one antibody against SARS-CoV-2 in the NegV group is comparable with other reported percentages, ranging anywhere from 0.2% to 24% [23]. In Central Europe, Eastern Europe, and Central Asia, the median seroprevalence was at 2.83% [24], considerably lower than the seroprevalence detected in our NegV group. This only furthers the fact that the true spread of the virus is underreported, but does not undermine the steady increase in natural herd immunity across the Cypriot community.

SARS-CoV-2 NP and S1RBD-specific antibody responses were monitored over time. Our results show that anti-NP IgG antibodies decreased with time, similar to what has been previously reported [25,26,27,28,29]. This further confirms the importance of the role of anti-NP IgG antibodies during early stages of infection and not as long-lived contributors to immunological memory. In fact, the detection of anti-NP antibodies is more sensitive than that of anti-S1RBD antibodies during early stages of infection, thus allowing for better and more effective serological diagnosis of the infection during its early stages [30,31]. On the contrary, and considering their role as neutralizing antibodies, anti-S1RBD IgG antibodies were relatively stable across time, and although they showed a significant drop within the first few months from the infection, their levels plateaued beyond that point. Other studies have reported similar results [25,26,27,28,29], thus confirming the stability of anti-S1RBD IgG antibodies and their importance in immunological memory. Additional analysis was performed by visualizing anti-NP and anti-S1RBD IgG level change across time. For this analysis, the antibody levels from the PosV group comprising the 695 volunteers were graphed against time, represented by days elapsed between sampling and SARS-CoV-2 infection confirmation (Appendix A). A significant decrease in anti-NP IgG (linear regression analysis; *p* = 0.02) and a stable level in anti-S1RBD IgG (*p* = 0.4) further reinforce our results above.

The study faced multiple limitations, one of which was the nature of the sampled population. Volunteers were mostly restricted to the capital city of Nicosia due to convenience in providing blood samples. Therefore, extrapolation of our results for the whole island should be done with caution. As this was the first study of its kind in Cyprus, further studies can focus on seroprevalence in the general population by considering a more inclusive volunteering plan. Additionally, utilizing more than one test/kit in future experiments can aid in determining the validity of the status of the samples for SARS-CoV-2 antibodies. Finally, there exists a limitation concerning the discrepancy in the time points at which the samples were collected; however, considering the volunteering element in the study design, great care was taken to not skew our data from representing a reasonable depiction of the situation.

## 5. Conclusions

Overall, this study shows that at least 14.4% of the general population has been naturally infected with SARS-CoV-2 without having a confirmed RT-PCR test. Of equal importance, we show that the antibody response, specifically against S1RBD, following natural infection persists for at least six months, highlighting, like other studies, that perhaps recovering COVID-19 patients should receive only a single shot of vaccine, which can act as a booster months after infection [26,32,33].

## Figures and Tables

**Figure 1 microorganisms-10-00085-f001:**
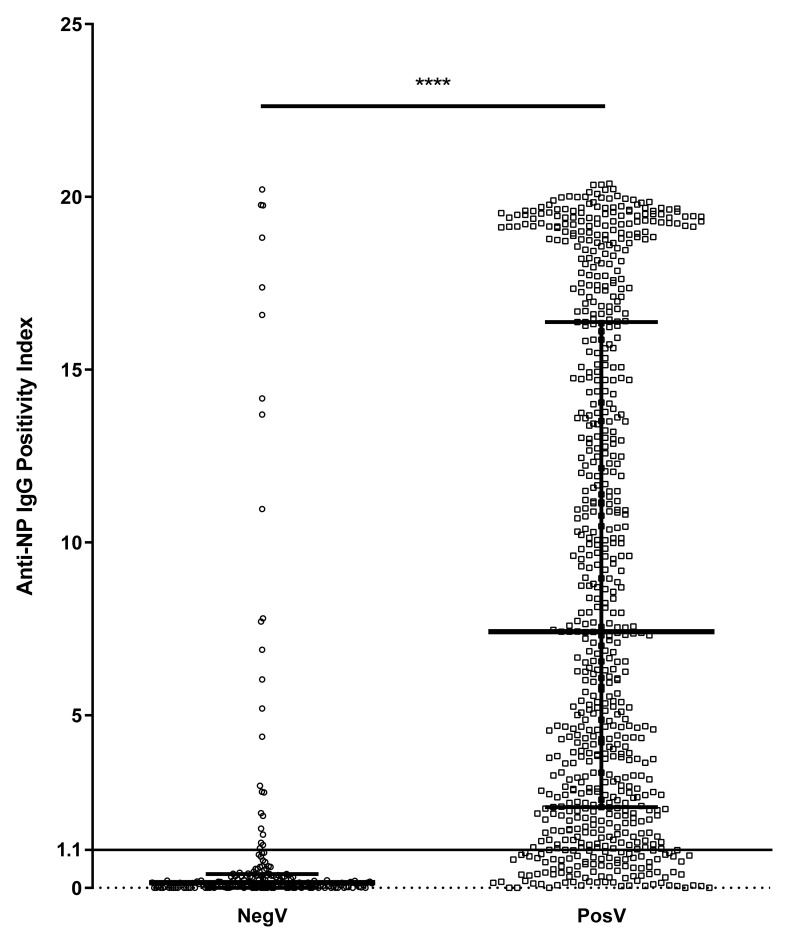
Distribution of positivity index (P.I.) of anti-NP IgG antibodies between negative/untested volunteers (NegV) (*n* = 194) and SARS-CoV-2-positive volunteers (PosV) (*n* = 695). The line at P.I. = 1.1 represents the threshold above which samples are considered positive. Bars represent median and interquartile ranges. **** *p* < 0.0001.

**Figure 2 microorganisms-10-00085-f002:**
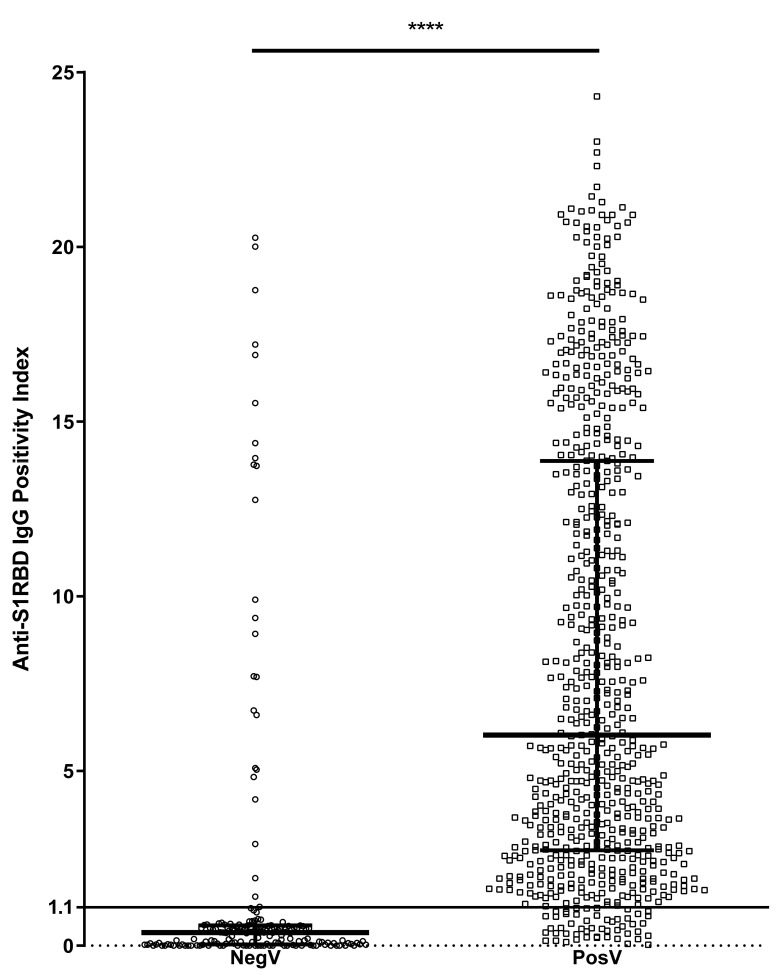
Distribution of positivity index (P.I.) of anti-S1RBD IgG antibodies between negative/untested volunteers (NegV) (*n* = 194) and SARS-CoV-2-positive volunteers (PosV) (*n* = 695). The line at P.I. = 1.1 represents the threshold above which samples are considered positive. Bars represent median and interquartile ranges. **** *p* < 0.0001.

**Figure 3 microorganisms-10-00085-f003:**
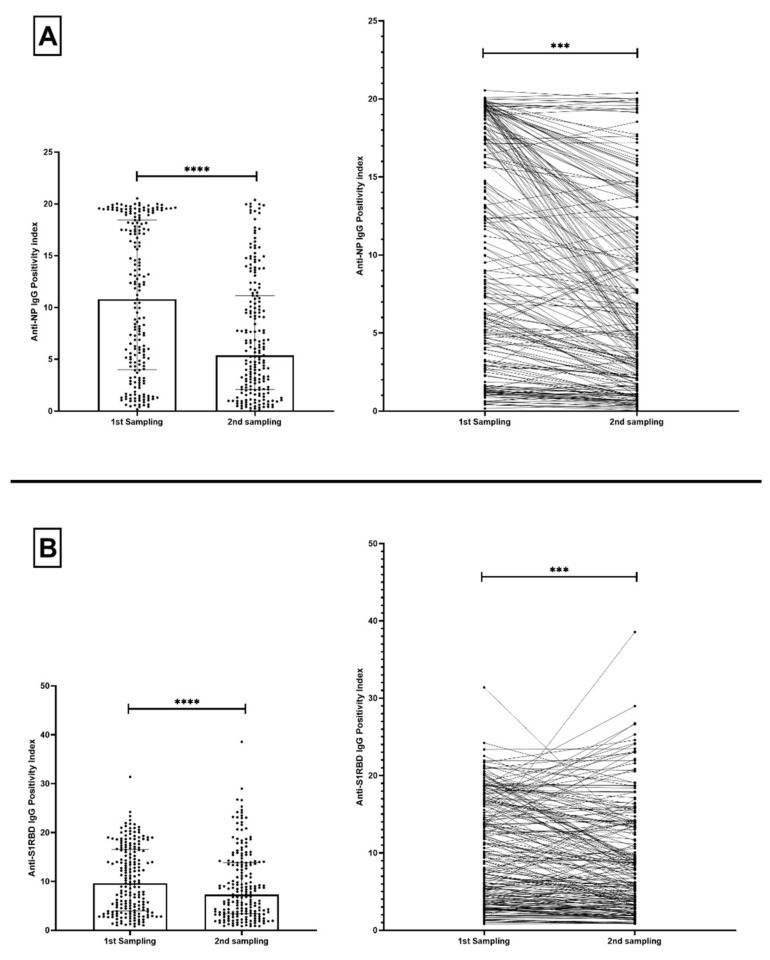
Distribution (**left**) and correlation graph (**right**) of positivity index (P.I.) of (**A**) anti-NP IgG and (**B**) anti-S1RBD IgG antibodies across 2 sampling points of antibody-positive volunteers (*n* = 205). Bars represent median and interquartile ranges. *** *p*<0.001; **** *p* < 0.0001.

**Figure 4 microorganisms-10-00085-f004:**
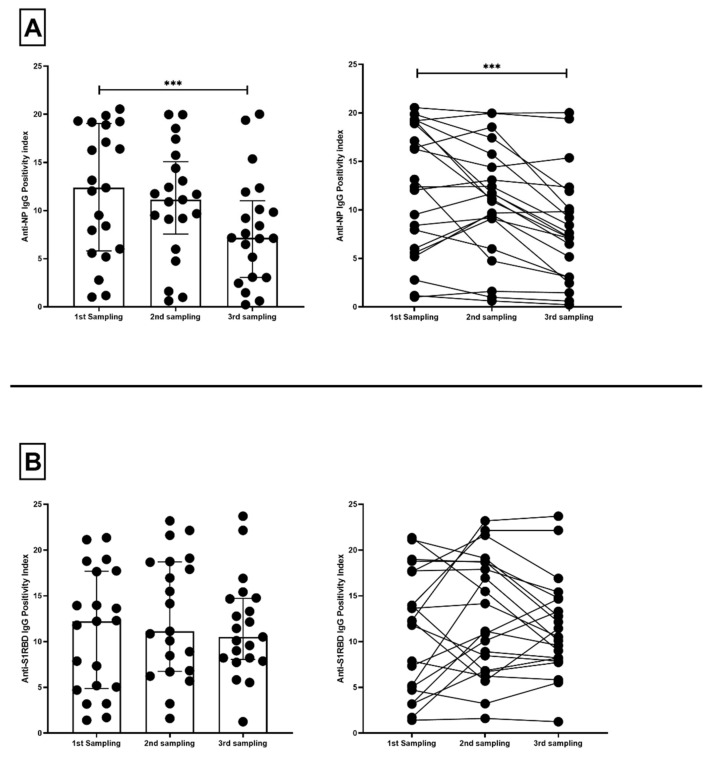
Distribution (**left**) and correlation graph (**right**) of positivity index (P.I.) of (**A**) anti-NP IgG and (**B**) anti-S1RBD IgG antibodies across 3 sampling points of antibody-positive volunteers (*n* = 21). Bars represent median and interquartile ranges. *** *p* < 0.0001.

**Table 1 microorganisms-10-00085-t001:** Characteristics of study participants. The Mann–Whitney U test was used for age matching, and the Fisher’s exact test was used for gender matching.

Features	SARS-CoV-2-Positive Volunteers (PosV)	SARS-CoV-2-Negative/-Unknown SARS-CoV-2 History (NegV)	*p*-Value
Number of volunteers	695	194	
Age (mean (SD))	47.95 ± 13.92	45.92 ± 14.57	0.051
Sex (male/female)	283/412	85/109	0.459

**Table 2 microorganisms-10-00085-t002:** Crude and adjusted prevalence of NP and S1RBD IgG in volunteers who tested negative/were unknown (NegV) (*n* = 194) and those who tested positive (PosV) (*n* = 695) for SARS-CoV-2. Adjustment calculations are based on the sensitivity and specificity of the kits used as provided by the manufacturer.

Volunteers	*n*	Positive for Anti-NP IgG	Crude Prevalence	Adjusted Prevalence	Positive for Anti-S1RBD IgG	Crude Prevalence	Adjusted Prevalence
PosV	695	598	86.04%	88.59%	645	92.81%	100%
NegV	194	25	12.89%	12.38%	24	12.37%	6.61%

## Data Availability

The data presented in this study are available on request from the corresponding author. The data are not publicly available due to privacy and ethical reasons.

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
