# Peer review of "Characterization of IgG Antibody Response against SARS-CoV-2 (COVID-19) in the Cypriot Population"

_microorganisms, 2021, doi:10.3390/microorganisms10010085_

Round 1
Reviewer 1 Report
The purpose of the manuscript is to evaluate the level of IgG antibodies including anti-NP and anti-S1RBD antibody against SARS-CoV-2 (Covid-19) in the Cypriot population. The results demonstrated most of the infected people developed the high level of antibodies against the virus after they were reported positive by RT-PCR. The level of anti-NP antibodies decreased more significantly than anti-S1RBD one across time. In addition, the percentage of people regarded as negative and undetermined samples possessing at least one antibody was comparable with world-wide data. This work contributed to the global database which lacks the information of SARS-CoV-2 serosurveillance in Cyprus. However, several issues need to be addressed prior to the publication.
- The manuscript reads like a report rather than a research article. The data should be more abundant and convincing.
- The time of the 1st sample collection ranges from 20 days to 6 months, thus the samples can be divided into 6 groups by month. Subsequent comparisons can be made for each group. Otherwise, the discrepancy among the groups was covered by the average value.
- To illustrate the changing of the level of antibodies across time, it is recommended to compare the changes among groups which can be divided by the different interval time. This means the independent variable should be the various time after the 1st (or 2nd) sample collection, while the dependent variable should be the difference of level change in corresponding the interval time. It will be more accurate than Figure 3 and 4 to show the variety of antibodies with time.
- The interval periods of each sample collection are unreasonable. The time range of sample collection in one group is too wide that it is almost equal to the interval among the 1st, 2nd and 3rd sample collections, leading to no comparability in some data and impairing the accuracy.
- In Figure1 and 2, the NegV on the X axis is not exact because the samples include the negative and unknown volunteers.
- Although the commercial ELISA kit used in the article, the sensitivity and specificity of each antibody should be confirmed and show the experimental data rather than introducing via the manufacturer.
- The whole article lacks novelty, new method, or new discovery for combating the virus?
- Why did the author not track those people who had at least one antibody against the virus in the NegV group subsequently? Did those people transfer into positive later? If not, the reason to explain why the percentage was high is invalid.
Reviewer 2 Report
the work is well done however the resonance of the work is limited, a shorter communication could be useful
Reviewer 3 Report
The authors Krashias et al present “Characterisation of IgG antibody response against SARS-CoV-2 (Covid-19) in the Cypriot population”, an article detailing the spread of SARS-CoV-2 on the island of Cyprus over the pandemic. The article is well written but some improvements are required before publication
General comments
- Experimentation: ELISA - Using a commercial kit increases replicability of the findings reported, however, the paper suffers from lack of the use of an international standard to standardize these finding to other laboratories. This should be noted in the discussion.
- Please could the authors consider mentioning the demographics of sampled patients, were these all in Greek administered Cyprus or across the whole city/island?
Minor corrections
Line 41- Prevalence of 86% seems high, but as this is a published number, it raises further questions that the authors should discuss in this manuscript. This suggests that the vast majority of Cypriots were infected in the first wave, yet still significant rates of infection were detected by B.1.1.7 and then B.1.617.2! It would be beneficial to the manuscript and the readers to discuss this, as this would increase the impact of the current study and its relevance to the audience.
Line 49- With a population below 1 million, the authors should consider a different way of describing cases on the island, or clarify that this means X cases in the total population to bring it into context.
Line 53- Replace “contagiousness” with “transmissibility”.
Line 58- 13th
Line 65 to 67- Please would the authors expand on the importance of these seroprevalence studies and the reasoning behind carrying out this study.
Line 67- 0.93% to 2.18% ?
Line 85- Please could the authors state the timeframe between PCR positive/negative and recruitment in this article
Line 87- Please could the authors clarify what information was gathered relating to previous history of COVID-19.
Line 104- The authors should expand the text to include whether re-testing for SARS-CoV-2 exposure was performed between time points.
Line 118 and 119- Please could the authors clarify what the negative control, positive control and calibrator were if possible. Clarify the exact kit used from ImmunoDiagnostics
Line 120- The authors should clarify what wavelength the absorbance was measured at, and further explain the P.I calculation for the reader – it is unclear how this is calculated, and why NP and S1RBD are calculated in different ways. Include more details on the basics of the ELISA performed, more ELISA details needed.
Line 144 onwards- Please could the authors include more information and justification for adjusted prevalence (for example adjusted prevalence of 100% for S1RBD IgG is alarming)
Line 145- 645/695 is 100%, is this a number typo or rounded up?
Line 148- Remain consistent with values presented and decimal places (95.4% and 14.4%, for example)
Line 159 (and other figures lines 163/173/183)- Ideally positive and negative control (from recognized sources) PI values should be incorporated into figures.
Line 173- The authors should consider linking dots from 1st and second samples to highlight the drop in antibody titre from 1st to 2nd sampling. This would increase the amount of data presented in these figures. This is an option in PRISM
Line 179- The authors have moved from reporting a % drop, to reporting raw P.I values, please adjust to only using one way of reporting to simplify the data for the reader, or ideally use both e.g. raw value and %
Line 184- Again, a different graph type linking individual samples across 3 time points would increase the impact of the figure.
Line 206- This is a broad statement about lockdown, that may not be necessary – as it is not possible to investigate this, I would suggest that the authors either remove this statement, or expand it to include other possible reasons (e.g. length of sampling period, different population dynamics, etc.)
Line 210- Please define the authors definition of herd immunity (and cite backing articles), especially in the context that the prevalence was 86% in the first wave, but still significant second and third waves occurred?
Line 213- This is a misleading sentence about the importance of NP antibodies, as the authors have not linked these antibody levels to the severity of disease found in the sampled individuals. Please could the authors cite an article detailing the importance of NP antibodies to viral clearance at the early stage of infection (as suggested) or modify this sentence.
Line 216- The S1 region is target to both neutralising and non-neutralising antibodies, please could the authors be careful when generalizing that S1 antibodies are neutralising, as suggested in this sentence.
Line 222- Please expand on the other limitations in regard to cohort sampling, particularly the large discrepancy in the sampling time elapsed from natural infection/RT-qPCR and sample collection between participants, as well as between consecutive bleeds for longitudinal analysis.
Line 227- Due to the sampling biases and errors, i.e. sampled from one city and large discrepancy in time elapsed between participants disease progression at the time of sampling, these results are only indicative of seroprevelence to SARS-CoV-2 endemic in Nicosia and not the wider general population, please could the authors factor this into the text to avoid sweeping generalisations.
Reviewer 4 Report
The authors present their study related to seroprevalence of SARS-CoV-2 antibodies in the Cypriot population. The manuscript is very well written and clearly presented.
I have some suggestions to increase the significance of this manuscript:
- The authors should make a comment on the reason why IgG antibodies for NP protein and S1 protein where found in different proportions among the analyzed samples.
- The authors should also relate the occurrence of IgG Abs in blood circulation with the time (take into consideration the time of infection as well as the sequencing samples that were available), as well as the possible lifetime of this Abs as some people that have infected by COVID-19 had undetectable Abs.
Round 2
Reviewer 1 Report
The author answer and revise most of the questions, can be publish as it is.